# Molecular mechanism of complement inhibition by the trypanosome receptor ISG65

Alexander D Cook[1,2], Mark Carrington[3]*, Matthew K Higgins[1,2]*

[1]Department of Biochemistry, University of Oxford, Oxford, United Kingdom; [2]Kavli Institute for Nanoscience Discovery, Dorothy Crowfoot Hodgkin Building, University of Oxford, Oxford, United Kingdom; [3]Department of Biochemistry, University of Cambridge, Cambridge, United Kingdom

**\*For correspondence:**
mc115@cam.ac.uk (MC);
matthew.higgins@bioch.ox.ac.uk (MKH)

**Competing interest:** The authors declare that no competing interests exist.

**Abstract** African trypanosomes replicate within infected mammals where they are exposed to the complement system. This system centres around complement C3, which is present in a soluble form in serum but becomes covalently deposited onto the surfaces of pathogens after proteolytic cleavage to C3b. Membrane-associated C3b triggers different complement-mediated effectors which promote pathogen clearance. To counter complement-mediated clearance, African trypanosomes have a cell surface receptor, ISG65, which binds to C3b and which decreases the rate of trypanosome clearance in an infection model. However, the mechanism by which ISG65 reduces C3b function has not been determined. We reveal through cryogenic electron microscopy that ISG65 has two distinct binding sites for C3b, only one of which is available in C3 and C3d. We show that ISG65 does not block the formation of C3b or the function of the C3 convertase which catalyses the surface deposition of C3b. However, we show that ISG65 forms a specific conjugate with C3b, perhaps acting as a decoy. ISG65 also occludes the binding sites for complement receptors 2 and 3, which may disrupt recruitment of immune cells, including B cells, phagocytes, and granulocytes. This suggests that ISG65 protects trypanosomes by combining multiple approaches to dampen the complement cascade.

## eLife assessment

This **fundamental** study significantly advances our understanding of how parasites evade the host complement immune system. The new cryo-EM structure of the trypanosome receptor ISG65 bound to complement component C3b is highly **compelling** and well-supported by biochemical experiments. This work will be of broad interest to parasitologists, immunologists, and structural biologists.

## Introduction

African trypanosomes can survive in the blood and tissue spaces of mammals for decades (*Sudarshi et al., 2014*), despite constant exposure to the molecules and cells of the immune system. They have evolved a unique surface coat packed with many copies of a single variant surface glycoprotein (VSG) (*Schwede et al., 2015*). At a population level, antigenically distinct VSGs are expressed over the course of an infection, thereby preventing antibody-mediated clearance (*Schwede and Carrington, 2010*). In addition to the need to resist acquired immunity, trypanosomes must also evade innate immune processes, such as the complement system. Recent studies have identified receptors which function within the trypanosome surface coat and which bind to either complement factor C3b (*Macleod*

*et al., 2022*) or complement modulator factor H (*Macleod et al., 2020*). ISG65 was identified as the trypanosome C3b receptor and has been shown to reduce the susceptibility of trypanosomes to antibody-mediated clearance in a mouse infection model (*Macleod et al., 2022*). However, we have little insight into the molecular mechanisms underpinning complement resistance mediated by ISG65.

The complement system involves a complex set of molecular cascades (*Gros et al., 2008*; *Zipfel et al., 2013*). These come together at the conversion of serum complement C3 into C3b and the deposition of C3b on a pathogen surface through the formation of a thioester bond between the TED domain of C3b and cell surface components (*Ricklin et al., 2016*). C3b deposition can occur through three major and distinct pathways. In the classical pathway, antibodies mediate the recruitment of C3b, while in the lectin pathway, this results from the recognition of cell surface glycans. In both cases, these events establish C4bC2b convertases, which catalyse the conversion of C3 into C3b and its surface deposition. In contrast, the alternative pathway involves stochastic conversion of C3 into C3b, resulting in an initial deposition event independent of other molecular recognition processes (*Nilsson and Nilsson Ekdahl, 2012*). The first deposited C3b molecules can then assemble with factors B and D, leading to formation of the C3 convertase, C3bBb, which catalyses deposition of further C3b molecules and amplification of downstream responses (*Rooijakkers et al., 2009*).

The outcomes of C3b deposition are also diverse, involving both the cellular and molecular branches of the immune system. Direct recognition of immobilised C3b, or its cleavage products iC3b, C3dg, and C3d, by complement receptors stimulates the activity of various immune cells. Complement receptor 1 (CR1) is found on macrophages and binding of CR1 to C3b promotes phagocytosis of pathogens such as *Leishmania* (*Rosenthal et al., 1996*). Complement receptor 2 (CR2) is found on B cells and forms a signal-transducing B cell co-receptor with CD19 and CD81 (*Bradbury et al., 1992*). CR2-CD19-CD81 is stimulated upon binding to C3d, and the absence of CR2 severely attenuates humoral immunity (*Fischer et al., 1998*; *Croix et al., 1996*). Complement receptors 3 and 4 are integrins found on various leukocytes and are associated with diverse effects, such as enhancement of natural killer cell cytotoxicity and antibody-dependent eosinophil cytotoxicity against schistosomes (*Erdei et al., 2019*; *Capron et al., 1987*). Through mechanisms distinct from those mediated by complement receptors, C3b can trigger a cascade which leads to recruitment of the pore-forming membrane attack complex (*Tegla et al., 2011*). Here, deposited C3b binds to other complement factors, resulting in formation of a C5 convertase. This cleaves complement factor C5, generating C5b, which recruits factors C6 and C7 to cause membrane association. Factors C8 and C9 can then bind to C5b7 on the pathogen surface, leading to the formation of a pore which mediates cell death (*Couves et al., 2023*).

Pathogens have evolved a wide range of different approaches to evade complement-mediated destruction by regulating different stages of the complement cascade (*Zipfel et al., 2013*; *Lambris et al., 2008*). These include *S. aureus* Efb-C which binds to the TED domain of C3 and prevents the conformational change required to generate C3b *Hammel et al., 2007*; *S. aureus* Efb, Ehp, and Sbi which bind to the TED domain of C3/C3d and prevent binding of complement receptor 2, thereby inhibiting B cell recruitment *Ricklin et al., 2008*; *Isenman et al., 2010*; smallpox virus SPICE which displaces factor B, preventing C3 convertase function *Forneris et al., 2010*; and *S. aureus* SCIN and Sbi which bind to the C3bBb C3 convertase and hold it in an inactive conformation (*Rooijakkers et al., 2009*; *Clark et al., 2011*). This, therefore, raised the question of how ISG65 regulates complement-mediated processes. Does it inhibit the deposition of C3b by preventing the function of C3 convertases? Does it block the recognition of C3b by complement receptors, thereby reducing recruitment of immune cells? Does it block the function of the C5 convertase, preventing formation of the membrane attack complex? Here, we combine structural biology and biophysical methods to show that ISG65 does not block C3 convertase formation, but instead may combine multiple functionalities to dampen the outcomes of C3b deposition.

## Results
### Two distinct binding sites connect C3b to ISG65
We previously determined the crystal structure of ISG65 bound to C3d (equivalent to the TED domain of C3b), revealing how the three core helices of ISG65 form a concave surface to which C3d binds (*Macleod et al., 2022*). However, this study also showed that this structure does not reveal the full

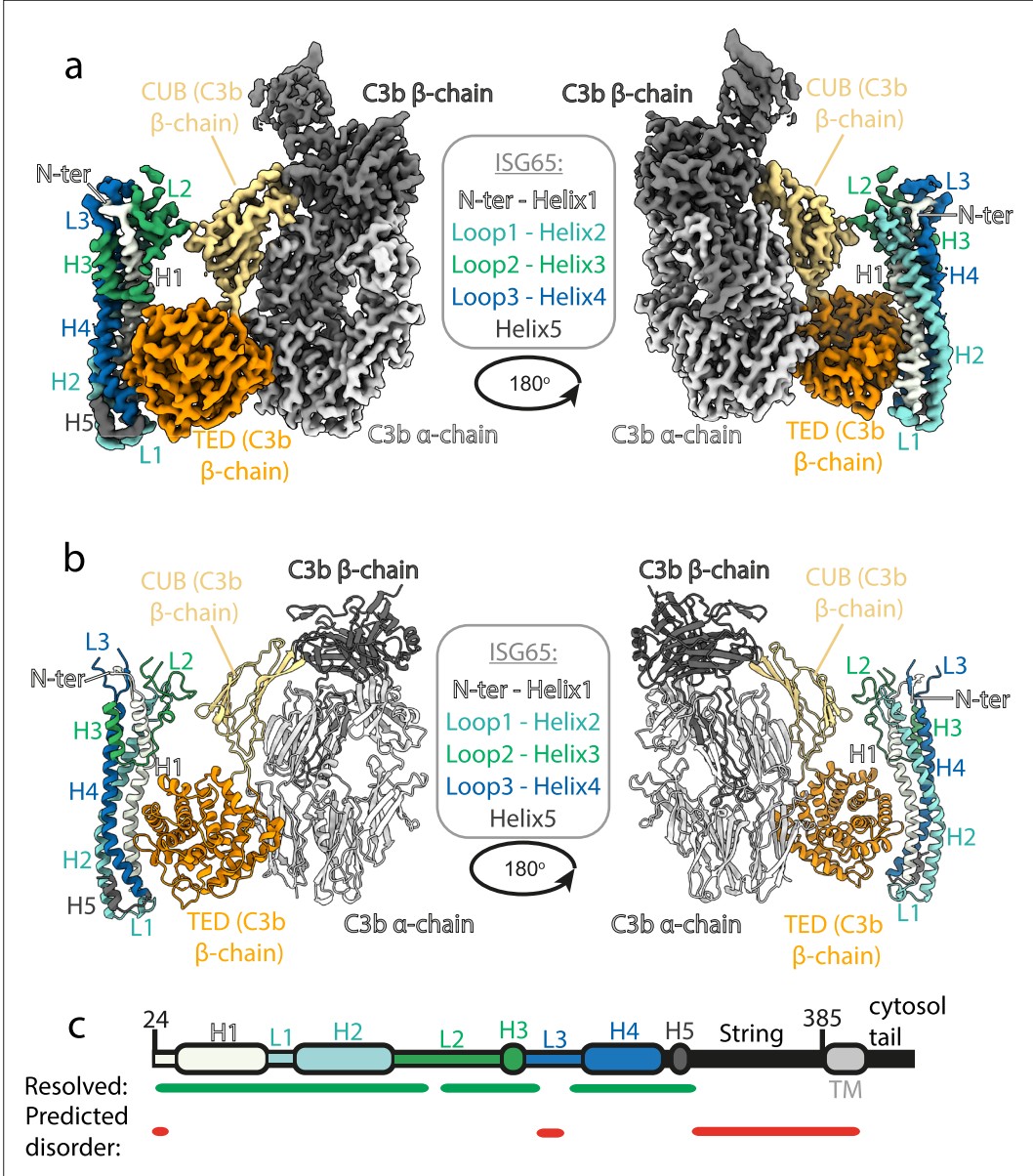

**Figure 1.** Structure of the complex between ISG65 and human C3b. (**a**) Composite volume of locally refined regions determined using cryogenic electron microscopy for ISG65 bound to human C3b. ISG65 is coloured in different shades of blue and green, as indicated in the legend in the centre of the panel (loop1 and helix 2 are light blue, loop 2 and helix 3 are green and loop 3 and helix 4 are dark blue). C3b is coloured in grey scale with the α-chain in light grey and the β-chain in dark grey. The TED domain is highlighted in orange and the CUB domain highlighted in yellow. (**b**) Molecular model of the same complex with a colour scheme matching that of (**a**). (**c**) A schematic showing the features of ISG65, coloured as (**a**). Regions resolved in the structure are indicated underneath the schematic using a green line and regions predicted to be disordered using AUCpreD (**Wang et al., 2016**) are shown by the red line.

The online version of this article includes the following figure supplement(s) for figure 1:

**Figure supplement 1.** Workflow of Cryo-EM data processing.

interaction interface between ISG65 and C3b. Surface plasmon resonance had been used to measure the affinities of ISG65 for the different fragments of C3, C3b, and C3d (**Macleod et al., 2022**). C3b exhibited a higher affinity for C3b than C3d, suggesting that ISG65 forms contacts with C3b in addition to those structurally characterised with the TED domain.

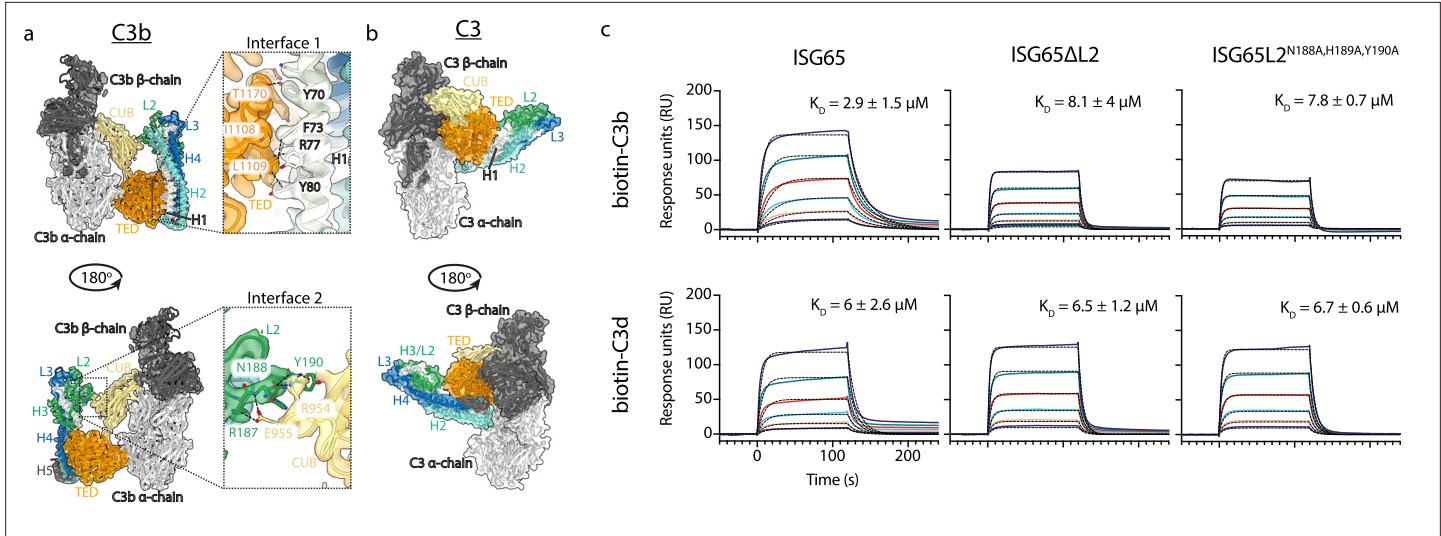

**Figure 2.** ISG65 forms two distinct interfaces with the TED and CUB domains of C3b. (**a**) The ISG65-C3b model shown in transparent cryo-EM density. The top panel shows the interface between ISG65 and the TED domain (orange), with bottom panel showing the interface between loop L2 of ISG65 (green) and the CUB domain of C3b (yellow). In each case, the left-hand panel shows the intact structure, with a dotted box highlighting the region shown in an enlarged form in the right-hand panel. (**b**) The ISG65 model superimposed onto a previously determined structure of C3 (PDB ID: 2A73) (*Janssen et al., 2005*) via the TED domain of the ISG65-C3b model. This is shown as a ribbon within a transparent surface representation. ISG65 can bind to C3 via the TED domain, via the same interface as previously identified for ISG65-C3d (*Macleod et al., 2022*). (**c**) Surface plasmon resonance data showing responses from the injection of ISG65, ISG65ΔL2, and ISG65$^{N188A,H189A,Y190A}$ (twofold serial dilutions from a concentration of 10 µM) over a flow cell coupled to biotin-C3b or biotin-C3d. Data is representative of three experimental repeats. Raw data is available in *Figure 2—source data 1*.

The online version of this article includes the following source data and figure supplement(s) for figure 2:

**Source data 1.** Surface plasmon resonance data.

**Figure supplement 1.** Surface plasmon resonance data.

**Figure supplement 1—source data 1.** Surface plasmon resonance data.

**Figure supplement 1—source data 2.** Surface plasmon resonance data.

To provide a full molecular model of ISG65 bound to C3b we used cryogenic electron microscopy (*Figure 1*). We prepared ISG65-C3b complex in the presence of fluorinated octyl maltoside, which improved particle distribution in grids while avoiding dissociation of the complex. We collected 14,339 movies from which particles were extracted and a three-dimensional volume was calculated. To improve the resolution of the region containing the binding site, local refinement was performed using a mask covering ISG65 and the TED and CUB domains of C3b, resulting in a volume at 3.4 Å resolution. Guided by previous structures of ISG65 (*Macleod et al., 2022*) and C3b (*Janssen et al., 2006*) and by an Alphafold2 (*Jumper et al., 2021*) model of ISG65, we were able to build a molecular model for the ISG65-C3b complex (*Figure 1*, *Figure 1—figure supplement 1*, *Supplementary file 1*).

This structure reveals the two distinct interfaces formed between ISG65 and C3b (*Figure 1*, *Figure 2a*). The first of these, interface 1, matches that previously identified through our crystallographic analysis (*Macleod et al., 2022*), with no significant differences between the models in this region. While our previous structure did not have interpretable electron density for loops L2 and L3, perhaps due to their disorder, or due to proteolysis during crystallisation, most of L2 and parts of L3 were ordered and resolved in our cryogenic electron microscopy-derived volume. This allowed us to build a de novo model for residues 179–212 of L2. In particular, L2 directly contacts the CUB domain of C3b, with an electrostatic interaction centered around C3b residue Arg954. Docking suggests that this second interface does not form between ISG65 and C3, as also seen in a recent structure of ISG65 bound to C3b (*Sülzen et al., 2023*; *Figure 2b*). The presence of this additional contact between ISG65 and C3b, which is not present between ISG65 and the TED domain alone, explains the differences in affinity of ISG65 for C3, C3b, and C3d.

We, and others, had previously used surface plasmon resonance analysis to measure the binding of C3, C3b, and C3d to immobilised biotinylated ISG65 (*Macleod et al., 2022*; *Sülzen et al., 2023*;

*Lorentzen et al., 2023*). However, we were concerned that differences in size and shape between the C3 variants might cause them to interact differently in this assay due to differences in hydrodynamic properties affecting their on-rates. To reliably compare ISG65 variants, we, therefore, changed to an assay in which C3b and C3d were conjugated to the chip surface, allowing us to flow the same ISG65 samples over these surfaces. To conjugate C3b and C3d in a manner which closely matches their orientation when conjugated to a pathogen, we chemically biotinylated Cys1010 and captured it on a streptavidin-coated chip. ISG65 was flowed over immobilised biotinylated C3b and biotinylated C3d, showing binding which fitted a one-to-one binding model with an affinity of 2.9 μM for C3b, and 6 μM for C3d (*Figure 2*, *Figure 2—figure supplement 1*). As C3d contains all determinants for formation of interface 1, we attribute the greater affinity for C3b over C3d to the contacts formed in interface 2. Indeed, we next generated two mutant forms of ISG65 in which we aimed to disrupt interface 2, either through deletion of loop 2 (ISG65ΔL2), or through mutation of the three ISG65 residues in loop 2 which mediate interface 2 (ISG65L2$^{N188A,H189A,Y190A}$). In neither case did these mutations affect the affinity for C3d but both mutations reduced the affinity for C3b to match that for C3d, supporting the model in which interface 2 forms with C3b but not C3d (*Figure 2*, *Figure 2—figure supplement 1*).

## ISG65 does not inhibit formation of the C3 convertase but does form a specific covalent conjugate with C3b

In addition to determining the structure of C3b bound to ISG65, the same data set also yielded a three-dimensional class consisting of a structure of C3b which lacked density for ISG65 and was indistinguishable from previous C3b structures. This allowed us to determine whether the presence of ISG65 caused a conformational change in C3b (*Figure 3a*). Fitting the model of the C3b-ISG65 complex (without ISG65) into the volume derived for the complex resulted in a map-model correlation of 0.79. When we fitted the same model into the volume derived from C3b alone, the correlation was 0.76, indicating that the ISG65-bound conformation of C3b is equivalent to the free conformation of C3b. Therefore, unlike bacterial C3b-effector proteins, such as Efb-C (*Hammel et al., 2007*), ISG65 does not prevent C3 from adopting the active conformation of C3b. Indeed, this is consistent with ISG65 binding to C3b that is already conjugated to the trypanosome surface, rather than preventing C3b formation.

The initial conjugation of C3b to the trypanosome surface is followed by formation of the C3 convertase, consisting of C3b bound to factor Bb (C3bBb). This requires factor B to first bind to C3b and then be cleaved by factor D to generate C3bBb. In order to determine whether ISG65 can block C3bBb formation, we first compared the ISG65-C3b structure with those of C3b bound to factors B and D[30]. This indicates that ISG65 does not compete with either factor B or Factor D and does not block the binding of factor B (*Figure 3b*). This suggests that the C3 convertase can form in the presence of ISG65.

We, therefore, developed an in vitro assay for C3 convertase formation in which we combined C3 and factor B with catalytic quantities of C3b and factor D. When mixed in vitro, this triggered the cleavage of C3 to C3b, as shown by the production of C3a. In addition, it resulted in the cleavage of factor B to form Bb and Ba (*Figure 3c*). When performed with addition of a greater than three-fold excess of ISG65, the production of C3a and Ba were unaltered, indicating that formation of the C3bBb C3 convertase can proceed in the presence of ISG65. (*Figure 3c*). Indeed, two other recent reports also indicate that ISG65 does not affect formation of the C3 convertase (*Sülzen et al., 2023*; *Lorentzen et al., 2023*).

Comparison of the outcome of C3 convertase formation in the presence and absence of ISG65, revealed that the presence of ISG65 resulted in a high molecular weight band, which we identified through mass spectrometry to be a conjugate of ISG65 with C3b (*Figure 3c*, *Supplementary file 3*). When we conducted the equivalent experiment using the same amount of bovine serum albumin instead of ISG65, we did not observe the formation of this conjugate, suggesting that it occurs specifically due to the proximity of ISG65 and the thioester-forming residue of C3b when in the complex (*Figure 3d*). Finally, to identify which region of ISG65 is responsible for the formation of this conjugate, we used versions of ISG65 which lack loops L1, L2, or L3, or which lacked the flexible C-terminal region (ΔC). In each of the loop mutants, we still observed the formation of the ISG65-C3b conjugate. However, this was not observed in the ΔC mutant (*Figure 3e*). This C-terminal region is an unstructured string of 72 amino acids that does not form part of the binding site for C3b and is not

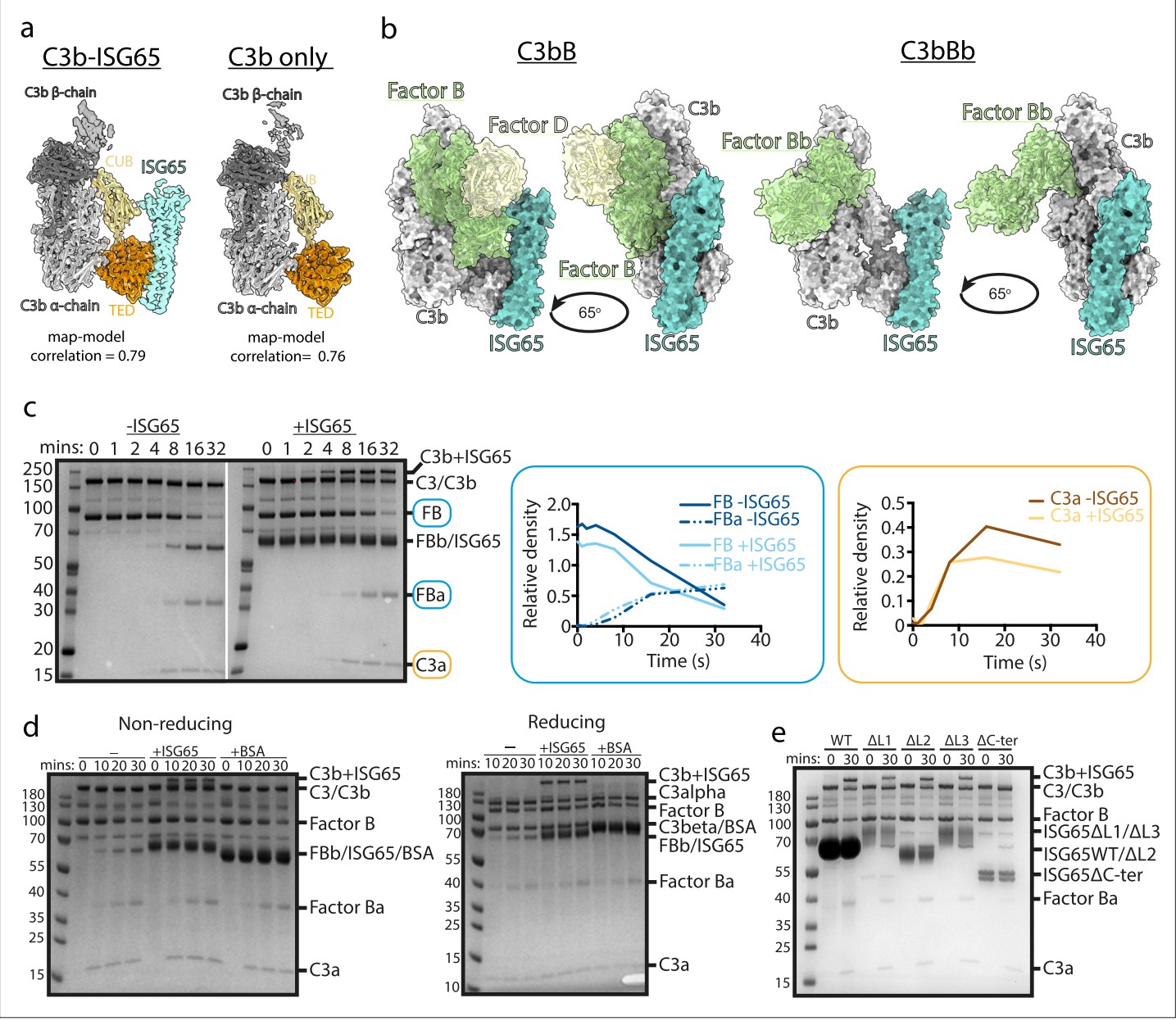

**Figure 3.** ISG65 does not block the formation of the C3 convertase. (**a**) The structure of the ISG65-C3b complex (without ISG65) docked into the electron microscopy-derived volumes obtained for the ISG65-C3b complex (left) and C3b alone (right). (**b**) Composite models obtained by docking the C3b-ISG65 structure onto those of C3b bound to factors B and D (PDB ID: 2XWJ) (***Forneris et al., 2010***) or factor Bb (6RUR) (***Rooijakkers et al., 2009***). (**c**) An assay for C3 convertase formation in which C3b and factor D were each added at concentrations of 12 nM and C3 and factor B at concentrations of 600 nM. Samples were taken at different time points and were analysed by SDS-PAGE analysis with Coomassie straining. This was done in the absence (left-hand gel) and presence (right-hand gel) of 2 µM ISG65. The graphs show quantification by densitometry for factors B, Ba and C3a to assess convertase function. (**d**) An equivalent assay to that shown in (**c**), conducted in the absence of non-complement protein (left), or the presence of 2 µM ISG65 (central) or 2 µM BSA (right). The left-hand gel was run in non-reducing conditions while the right-hand gel was run in reducing conditions. (**e**). An equivalent assay to that shown in (**c**), conducted in presence of 2 µM ISG65 or of ISG65 variants lacking loop 1 (ΔL1), loop 2 (ΔL2), loop 3 (ΔL3) or the extended disordered C-terminal region (ΔC-ter). Raw data available ***Figure 3—source data 1***.

The online version of this article includes the following source data for figure 3:

**Source data 1.** *Figure 3c* – raw gel 1 annotated.

**Source data 2.** *Figure 3c* – raw gel 1.

**Source data 3.** *Figure 3c* – raw gel 2 annotated.

**Source data 4.** *Figure 3c* – raw gel 2.

*Figure 3 continued on next page*

*Figure 3 continued*

**Source data 5.** *Figure 3d* – raw gel annotated.

**Source data 6.** *Figure 3d* – raw gel.

**Source data 7.** *Figure 3e* – raw gel annotated.

**Source data 8.** *Figure 3e* – raw gel.

observed in the structures. It is predicted to form a flexible linker which connects the structured ISG65 domain to the plasma membrane. These data, therefore, suggest that the proximity of the flexible linker of ISG65 to the thioester site of C3b, which occurs due to the interaction of ISG65 with C3/C3b, increases the likelihood of the thioester domain coming into contact with the ISG65 C-terminal linker, leading to the formation of a preferential conjugate between ISG65 and C3b. Indeed, as ISG65 can interact with C3 before conversion to C3b generates the reactive thioester, this conjugate may be preferred over conjugation of C3b to VSG, acting as a decoy to reduce the conjugation of C3b to other regions of the trypanosome surface. Whether this occurs on a trypanosome surface requires further experimentation.

## ISG65 blocks the binding of complement receptors 2 and 3 to C3b and C3d

As the central component of the complement system, C3 is the target of many host-proteins (*Ricklin et al., 2016*). These factors can be broadly grouped into the complement receptors, which are found on immune cells and bind to C3b, iC3b, C3db, and C3d fragments, and factors that regulate the activity of C3b. Complement regulators typically act by blocking recognition of C3b by host-factors to prevent downstream activation (*Noris and Remuzzi, 2013*). To test whether ISG65 might influence the capacity of complement regulators and receptors to bind to C3b/d, we next compared the structure of ISG65-bound C3b with previously determined structures of C3b and C3d bound to different complement regulators and receptors (*Figure 4*).

The conformation and location of ISG65 bound to C3d demonstrates that ISG65 binding would preclude binding of Factor H domains 19–20 (*Morgan et al., 2011*; *Figure 4*). In addition, ISG65 is predicted to have different effects on binding of complement receptors to C3b. The binding site on C3b for ISG65 does not overlap with those for C3b-binding complement receptors CRIg (*Wiesmann et al., 2006*) and CR1 (*Forneris et al., 2016*). However, the region on C3d occupied by ISG65 overlaps with sites on the TED domain/C3d which bind complement receptors 2 (CR2) (*van den Elsen and Isenman, 2011*) and 3 (CR3) (*Bajic et al., 2013*; *Figure 4*). CR2 is a receptor found on B cells, which in

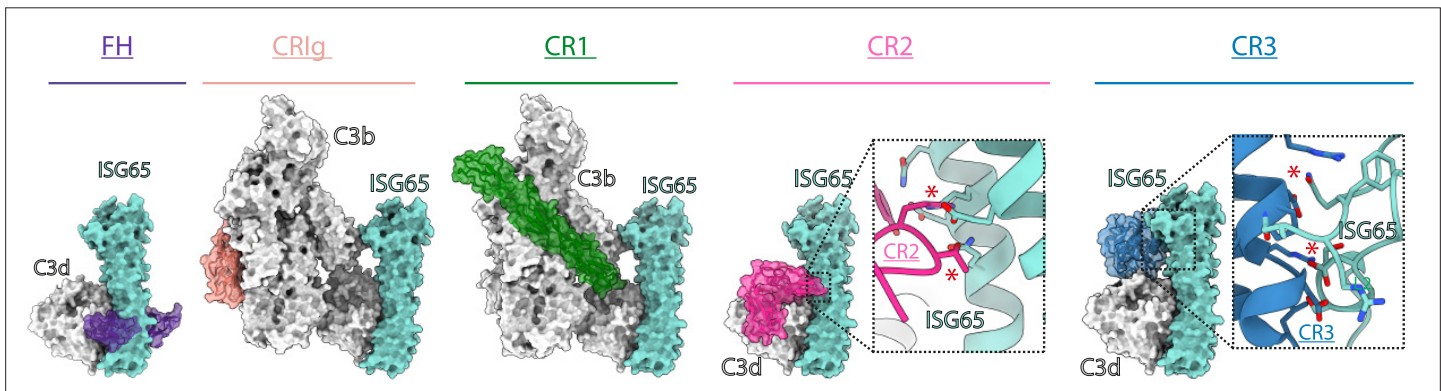

**Figure 4.** ISG65 overlaps the binding sites for complement receptors 2 and 3. Composite models obtained by docking the C3b-ISG65 structure onto those of C3b/d bound to factor H CCP19-20 (3OXU) (*Morgan et al., 2011*), CRIg (2ICF) (*Wiesmann et al., 2006*), CR1 CCP15-17 (5FO9) (*Forneris et al., 2016*), CR2 SCR1-2 (3OED) (*van den Elsen and Isenman, 2011*) and CR3 I-domain (4M76) (*Bajic et al., 2013*). C3b/d is shown in a solid light grey surface, ISG65 is shown in a solid turquoise surface, and complement regulators are shown in transparent surface with ribbon in various colours.

The online version of this article includes the following figure supplement(s) for figure 4:

**Figure supplement 1.** Comparison of ISG65 sequences from *T. brucei* brucei (green labels), *T. brucei* rhodesiense (blue labels), and *T. brucei* gambiense (red labels).

complex with CD19 and CD81, forms a signal transducing B-cell co-receptor (*Bradbury et al., 1992*). Binding of C3d to CR2 greatly reduces the threshold for B cell activation, thereby triggering B cell activation and antibody production (*Croix et al., 1996*). By preventing VSG-conjugated C3b from binding to B cells through an interaction mediated by CR2, ISG65 may reduce the likelihood that C3b-conjugated trypanosomes will induce B-cell activation and antibody production. Similarly, the binding site for CR3 on C3d also overlaps with that for ISG65, suggesting that ISG65 will block CR3 binding. CR3 is widely expressed on various immune cells and is known to promote macrophage recruitment and phagocytosis by binding to iC3b/C3d, indicating that ISG65 may help reduce trypanosome clearance by blocking this interaction (*Erdei et al., 2019*).

## Discussion

The long-term survival of a pathogen in a mammalian host can only occur if it has evolved strategies to avoid clearance by all arms of the host immune system, including the complement system. In a previous study, we highlighted the importance of the complement system in the clearance of trypanosomes during the first wave of infection in a mammalian infection model (*Macleod et al., 2022*). Mice infected with trypanosomes showed two waves of infection. The first peaked around five days after infection and was partially controlled. Around eight days after infection, a second wave was initiated, most likely due to trypanosomes which had undergone antigenic variation through switching their VSG coat. When a similar infection experiment was conducted using the same trypanosome cell line to infect mice lacking complement C3, then the first wave of infection was no longer controlled. This suggested that control of the first wave of infection was mediated by both antibodies and by complement, implicating the classical complement pathway. When wild-type mice were infected with trypanosomes lacking the complement C3/C3b receptor, ISG65, the control of the first wave of infection was delayed, suggesting that ISG65 reduces the susceptibility of trypanosomes to destruction by complement (*Macleod et al., 2022*). However, this study did not investigate the molecular mechanism by which ISG65 reduces the activity of complement.

Our previous structural studies revealed that ISG65 binds to C3d, which is equivalent to the isolated TED domain of C3b (*Macleod et al., 2022*). However, they also suggested that this does not describe the full interaction interface between ISG65 and C3b, with ISG65 showing a ~10 fold higher affinity for C3b than it shows for C3d (*Macleod et al., 2022*). To understand the molecular mechanism for ISG65 function, we therefore needed to reveal the full C3b binding mode of ISG65. We now show, through cryogenic electron microscopy, that in addition to interacting with the TED domain, ISG65 also interacts with the CUB domain of C3b, simultaneously bridging these two sites with a second structure, of C3b bound to ISG65 from *T. b. gambiense*, giving the same conclusion (*Sülzen et al., 2023*). Indeed, there are no consistent differences between the ISG65 receptors from *T. brucei* and its human-infective subspecies, *T. b. gambiense* and *T. b. rhodesiense* (*Figure 4—figure supplement 1*), making it highly likely that ISG65 functions in the same way in human and bovine-infective trypanosomes. This complete model of the ISG65-C3b complex now allows us to answer a series of questions about how ISG65 might modulate C3b function, showing whether ISG65 prevents the formation of C3b, whether it blocks formation of the C3 convertase and whether it blocks the binding of complement regulators and complement receptors to C3b.

The first conclusion is that ISG65 does not prevent the conformational changes which occur as C3 is converted to C3b, with no difference in conformation of free C3b and ISG65-bound C3b. Neither does ISG65 prevent the formation of the C3 convertase, C3bBb. This convertase forms when C3b recruits factors B and D, leading to cleavage of factor B to generate fragments Ba and Bb, with Bb remaining bound to C3b (*Gros et al., 2008*). The C3bBb convertase can then induce the formation of more C3b from C3, thereby increasing the quantity of surface-bound C3b and amplifying the complement cascade. ISG65 does not block the binding sites occupied by factors B or D, or the site proposed to be occupied by subsequent C3 molecules (*Rooijakkers et al., 2009*). Indeed, in a solution assay to measure C3 convertase function, we see that the presence of ISG65 has no effect on C3bBb activity. Indeed, two other reports, using different assays, also find no inhibition of C3 convertase formation by ISG65 (*Sülzen et al., 2023*; *Lorentzen et al., 2023*).

Intriguingly, while ISG65 does not affect C3 convertase function in this solution assay, we find that a newly formed conjugate is established between the flexible C-terminal tail of ISG65 and newly formed C3b. Indeed, the location of the C-terminal tail places it close to the thioester-forming residue in the

context of the ISG65-C3b complex. This conjugate is not formed when BSA is included in the assay at a similar concentration, or when ISG65 lacking the C-terminal tail is used. Could the formation of this conjugate help to protect the trypanosome from the downstream effects of C3b deposition on the cell surface? The amplification of C3b deposition, and the subsequent formation of the C5 convertase, requires C3b molecules and their binding partners to come into close proximity. It is possible that conjugating C3b to ISG65, which will swing above the trypanosome surface on a flexible linker, might make the C3b molecules less likely to come together productively than if they were linked to sites in the VSG surface.

Finally, our complete ISG65-C3b structure shows which binding sites for other complement receptors and regulators are occluded by the presence of ISG65. Indeed, we find that the binding sites on C3b and C3d for complement receptors 2 and 3 overlap with that of ISG65. These receptors are found on B cells and leukocytes, respectively. By blocking CR2 binding, ISG65 is likely to reduce B cell activation and antibody production, while blocking CR3 binding is likely to reduce trypanosome clearance by phagocytosis and complement-mediated cytotoxicity.

Therefore, our studies suggest that ISG65 might dampen the outcomes of the complement system through a diverse combination of mechanisms. By enhancing its affinity for C3b through a two-site binding mechanism, ISG65 will preferentially partition onto cell surface conjugated C3b than soluble C3. When ISG65 binds to C3 or C3b which is approaching the cell surface, a conjugate will preferentially be formed between the C-terminal tail of ISG65 and C3b, ensuring that C3b is flexibly attached rather than more rigidly associated with VSG, perhaps altering the likelihood of it forming productive complexes, such as C5 convertases. Finally, ISG65 may bind to VSG-conjugated C3b, blocking recruitment and stimulation of immune cells by the trypanosome surface, by preventing binding of CR2 and CR3. Other recent publications suggest that ISG65 might also inhibit formation of the C5 convertase (*Sülzen et al., 2023*) or accelerate the decay of C3b to iC3b (*Lorentzen et al., 2023*). Each of these effects could contribute to dampening of the complement response, while rapid clearance of surface attached C3b through hydrodynamic forces resulting from trypanosome swimming, coupled with rapid endocytosis, cleans the trypanosome surface.

While a number of functions have been ascribed to ISG65, it is noteworthy that none of the studies to date assess its function in the unusual context of a VSG-coated trypanosome surface, which may limit formation of the membrane attack complex and operation of the alternative pathway of complement (*Cook et al., 2023*). It will, therefore, be important for future studies to determine whether each of the functions proposed for ISG65, observed in in vitro assays, are also operational on trypanosomes before we can fully understand how ISG65 helps trypanosomes to survive.

# Methods

## Key resources table

| Reagent type (species) or resource | Designation | Source or reference | Identifiers | Additional information |
|---|---|---|---|---|
| Gene (*Trypanosoma brucei brucei*) | ISG65G gene | NCBI BioProject Accession: PRJEB46985 | UniProt: A0A8J9S0Z8 | |
| Gene (*Homo sapiens*) | CFB gene | NCBI GenBank accession: AF019413.1 | UniProt: P00751 | |
| Gene (*H. sapiens*) | CFD gene | NCBI GenBank accession: CH471139.2 | UniProt: P00746 | |
| Gene (*H. sapiens*) | C3 gene | NCBI GenBank accession: AY513239.1 | UniProt: P01024 | |
| Sequence-based reagent | Primers | This paper | See list of primers in the Appendeix. Primers were synthesised by Sigma. | |
| Recombinant DNA reagent | pHL-SEC vector backbone | https://doi.org/10.1107/S0907444906029799; *Aricescu et al., 2006* | | |
| Cell line (*H. sapiens*) | HEK293F | Gibco | R79007 | |
| Biological sample (*H. sapiens*) | Human serum | NHSBT non-clinical issue | | |
| Software | SIMPLE v3 | https://github.com/hael/SIMPLE/releases; *Elmlund et al., 2020* | | |

*Continued on next page*

*Continued*

| Reagent type (species) or resource | Designation | Source or reference | Identifiers | Additional information |
|---|---|---|---|---|
| Software | CryoSPARC v3 | https://cryosparc.com/; *Structura Biotechnology Inc, 2020* | | |
| Software | TOPAZ v0.2.4 | https://github.com/tbepler/topaz; *Bepler and Noble, 2020* | | |
| Software | RELION v3.1 | https://relion.readthedocs.io/en/release-3.1/index.html; *RELION developer team, 2020* | | |
| Software | DeepEMhancer | https://github.com/rsanchezgarc/deepEMhancer; *Sanchez Garcia, 2022* | | |
| Software | AlphaFold2 | https://github.com/google-deepmind/alphafold; *AlphaFold Team, 2021* | | |
| Software | ISOLDE v1.0 | https://github.com/tristanic/isolde; *Croll, 2019* | | |
| Software | COOT v0.9.8.3 | https://github.com/pemsley/coot; *Emsley, 2022* | | |
| Software | PHENIX v1.20.1 | https://phenix-online.org; *Phenix Development Group, 2022* | | |
| Software | ChimeraX v1.6 | https://www.cgl.ucsf.edu/chimerax/; *UCSF Resource for Biocomputing, Visualization, and Informatics, 2023* | | |
| Software | BIAevaluation v1.0 | Biacore, Cytiva, Marlborough, MA, USA | | |
| Software | Fiji | https://imagej.net/software/fiji/ | | |

## Mammalian expression and purification of ISG65 and complement proteins

To express ISG65 1125 G (*Macleod et al., 2022*) (residues 24–385), we used a pDest12 plasmid consisting of an N-terminal secretion signal, codon-optimized ISG65, a C-terminal flexible linker (GSGSGSASG), AviTag, and a His$_{10}$-tag. Human Complement Factor B (residues 26–764) and Complement Factor D (residues 20–253) were cloned into a pHLsec plasmid containing an N-terminal secretion signal and a short C-terminal linker (GSG) followed by a C-tag. ISG65, Factor B, and Factor D DNA were transfected into HEK293F cells (3 µg DNA per mL of cells) grown in F17 Freestyle media to a density of $2.2\times10^6$ cells/mL, using polyethylenimine (9 ug per mL of cells). Media was supplemented with 1 µM kifunensine and 3.8 mM valproic acid. Cell culture supernatant was harvested 6 days after transfection. Initial purification of ISG65 was performed using Ni Sepharose excel resin (Cytiva), whilst CaptureSelect C-tagXL Affinity Matrix (Thermo Fisher) was used to purify Factor B and D. ISG65 and Factor D were further purified on a Superdex 75 300/10 (Cytiva), whilst Factor B was further purified with a Superdex 200 300/10 (Cytiva). ISG65 loops deletions (loop1: ΔP88-K92insSS, loop2: ΔQ155-R195, loop3: ΔK230-P250, tail: ΔK317-G394), and ISG65$^{N188A,H189A,Y190A}$ were generated using Gibson Assembly (NEB) and expressed and purified as described for ISG65 24–385 above. ISG65 and ISG65ΔL2 were biotinylated on their C-terminal AviTag using the Enzymatic Biotinylated Kit (Sigma).

## Purification of human complement C3 and C3d, and generation of C3b

To purify Complement C3, anonymous donor post-clot human serum was obtained from the NHS Blood and Transplant non-clinical issue supply. Serum was buffer exchanged into 20 mM Tris pH 8, 50 mM NaCl, and 0.5 mM EDTA using tangential flow filtration with a stack of three 100 kDa Omega Cassettes (PALL Corporation). Serum was clarified by ultra-centrifugation at 41,000 rpm in a Ti-45 rotor (Beckman Coulter). Purification of C3 was performed by anion exchange chromatography using a HiPrep Q HP 16/60 column (Cytiva) with a 20-column volume gradient of 50–350 mM NaCl. Fractions containing C3 were pooled then buffer exchanged into 20 mM MES pH 6, 50 mM NaCl, and 0.5 mM EDTA using tangential flow filtration as above. C3 was then purified by cation exchange using a monoS 4.6/100 PE (Cytiva) with a 30-column volume gradient to 500 mM NaCl. Fractions containing C3 were then further purified on a Superdex 200 300/10.

C3b was generated from C3 by limited proteolysis with trypsin (Roche) at 1 % w/w trypsin to C3 at 37 °C for 2 min. Trypsin was then inhibited with soybean trypsin inhibitor (Merck) at a ratio of 1 % w/w inhibitor to C3. For biotinylation of C3b, 100 mM HEPES pH 7.0 was added after addition of soybean trypsin inhibitor, followed by a 10-fold molar excess of maleimide-PEG2-biotin (ThermoFisher). The reaction was incubated on ice for 6 hr. C3b or biotin-C3b was then purified on a Superdex 100 300/10.

We previously expressed C3d with a C1010A mutation to prevent formation of thioester bonds (*Macleod et al., 2022*). To generate C3d with a single biotin in proximity to the thioester-forming Gln$^{1013}$ residue, we generated a Q1013A mutation in C3d which prevented thioester bond formation but left Cys$^{1010}$ exposed. C3d$^{Q1013A}$ was expressed in *E. coli* as previous described for the C1010A mutant (*Macleod et al., 2022*) and was then reacted with maleimide-PEG2-biotin, as described above for C3b.

## Preparation of ISG65-C3b complexes for cryo-EM
To form C3b-ISG65 complexes, C3b was mixed with ISG65 at a 1:1.1 ratio in 20 mM HEPES pH 7.4, 150 mM NaCl, and 0.5 mM EDTA. Complexes were then purified on a Superdex 200 300/10 GL column. Quantifoil grids consisting of a 1.2/1.3 µm holey carbon film on 300 gold mesh were glow discharged at 15 mA for 1 min with an EM ACE200 glow discharger (Leica). Just before vitrification, 0.01% fluorinated octyl maltoside (Anatrace) was added to 2.2 mg/mL C3b-ISG65, which was then immediately added to the grid and plunge frozen in an ethane slush using a Vitrobot Mark IV (Thermo Fisher). Grids were imaged with a Titan Krios G2 (Thermo Fisher) operating at 300 kV, and images were recorded with a K3 detector (Gatan) in counting mode with a GIF Quantum LS Imaging Filter (Gatan).

## Image processing and modelling of ISG65-C3b complexes
Movies were motion-corrected, contrast transfer function (CTF) corrected, and particles were picked using SIMPLE v3 (*Caesar et al., 2020*) on the fly. To obtain an initial set of C3b/C3b-ISG65 particles, one round of 2D classification was performed in SIMPLE, followed by another two rounds of 2D classification in CryoSPARC v3 (*Punjani et al., 2017*). A second set of particles was obtained by particle picking with TOPAZ v0.2.4 *Bepler et al., 2019* followed by one round of 2D classification to remove bad particles. TOPAZ and SIMPLE particles were combined, duplicates removed, and a final round of 2D classification was performed. Three rounds of *ab initio* and heterogeneous 3D refinement were performed in CryoSPARC using 5 classes which resulted in a set of C3b-ISG65 particles, and a set of C3b only particles. Both particle sets were merged and yielded a 3.5 Å map from homogenous refinement in CryoSPARC. Bayesian polishing was then performed in Relion v3.1 (*Zivanov et al., 2018*; *Zivanov et al., 2019*), followed by per particle CTF refinement and beam tilt estimation in CryoSPARC, yielding a 3.3 Å map. Particles were separated into C3b-ISG65 and C3b only sets, yielding 3.3 and 3.4 Å resolution maps, respectively. The resolution of CUB, TED, and ISG65 were significantly lower than the rest of the map presumably because of flexibility in CUB and TED, and because of the location of ISG65 on the periphery of the map. To mitigate this, particle coordinates were shifted such that CUB-TED-ISG65 density was in the middle of the box, then all density other than CUB-TED-ISG65 was subtracted using a 10-pixel soft edge mask. Local refinement was then performed using a pose/shift Gaussian prior with a standard deviation of 3° over rotations and 2 Å over shifts, and search limitations of 12° and 9 Å, resulting in a 3.4 Å map. Local refinement was also performed for all density except CUB-C3d-ISG65 using a 15-pixel soft edge mask, yielding a 3.2 Å map. Post-processing was then performed using DeepEMhancer, and local resolution was estimated with CryoSPARC, and locally refined maps were combined in ChimeraX (*Pettersen et al., 2021*) to create a composite map.

To generate a model of ISG65-C3b, a previous crystal structure of C3b (PDB ID: 5FO7) (*Forneris et al., 2016*) and a structure prediction of ISG65 performed with AlphaFold2 (*Jumper et al., 2021*) were rigid-body fitted into cryo-EM density using the fit-in map tool in ChimeraX (*Pettersen et al., 2021*). Refinement of C3b-ISG65 was then performed using ISOLDE v1.0 (*Croll, 2018*), COOT v0.9.8.3 (*Emsley et al., 2010*), and Phenix v1.20.1 (*Liebschner et al., 2019*).

## Surface plasmon resonance
SPR experiments were performed on a BIAcore T200 (Cytiva). Biotin-C3b or biotin-C3d were immobilised on the SPR chip via streptavidin using a CAPture kit (Cytiva). Two-fold serial dilutions of ISG65,

ISG65ΔL2, and ISG65$^{N188A,H189A,Y190A}$ were injected over the chip. Measurements were performed at 30 μL/min at 25 °C in 20 mM HEPES pH 7.4, 150 mM NaCl, 0.05% TWEEN-20, with an association and dissociation time of 120 s. Binding responses were obtained using BIAevalutation software v1.0, followed by fitting to a 1:1 Langmuir model. Three experimental replicates of SPR experiments were performed, including two biological replicates of ISG65, biotin-C3d, and biotin-C3b.

## C3 convertase activity assays

To measure the effect of ISG65 on C3 convertase activity, 600 nM C3, 600 nM Factor B, 12 nM C3b, 12 nM Factor D, and 2 μM ISG65 or 2 μM bovine serum albumin (Sigma) were combined in phosphate-buffered saline pH 7.4, 2 mM MgCl$_2$. The reaction was carried out at 22 °C and samples were removed at various intervals and combined with SDS-PAGE sample buffer before running on SDS-PAGE to assess band shifts in C3 and Factor B. Gel densitometry was performed in Fiji (*Schindelin et al., 2012*).

## Acknowledgements

This work was funded through a Wellcome Investigator award (217138/Z/19/Z). We thank Olivia MacLeod for discussions about the complement system and its regulation and Rishi Matadeen, Joseph Caesar, and Teige Matthews-Palmer at the COSMIC cryo-EM facility (University of Oxford) for support with data collection and data processing.

## Additional information

### Funding

| Funder | Grant reference number | Author |
|---|---|---|
| Wellcome Trust | 217138/Z/19/Z | Alexander D Cook<br>Mark Carrington<br>Matthew K Higgins |

The funders had no role in study design, data collection and interpretation, or the decision to submit the work for publication. For the purpose of Open Access, the authors have applied a CC BY public copyright license to any Author Accepted Manuscript version arising from this submission.

### Author contributions

Alexander D Cook, Conceptualization, Data curation, Formal analysis, Investigation, Methodology, Writing - original draft, Writing - review and editing; Mark Carrington, Conceptualization, Formal analysis, Funding acquisition, Writing - original draft, Writing - review and editing; Matthew K Higgins, Conceptualization, Formal analysis, Supervision, Funding acquisition, Writing - original draft, Writing - review and editing

### Author ORCIDs

Mark Carrington (ID) http://orcid.org/0000-0002-6435-7266
Matthew K Higgins (ID) http://orcid.org/0000-0002-2870-1955

Reviewer #1 (Public Review): https://doi.org/10.7554/eLife.88960.3.sa1
Reviewer #3 (Public Review): https://doi.org/10.7554/eLife.88960.3.sa2
Author response https://doi.org/10.7554/eLife.88960.3.sa3

## Additional files

### Supplementary files

• Supplementary file 1. Supplementary Table 1: Cryo-EM data collection and model building statistics.

• Supplementary file 2. Supplementary Table 2: Kinetic parameters of ISG65 variants binding to biotinylated C3b and C3d, as measured by surface plasmon resonance. Average values and standard

deviation from three experimental repeats are reported.

• Supplementary file 3. Supplementary Table 3: Mass spectrometry analysis of ISG65-C3b conjugates observed in *Figure 3*. The high molecular weight band observed in convertase assay in the presence of ISG65 was excised and analysed by mass spectrometry. The three most abundant proteins identified are presented.

• Supplementary file 4. Primers.

• MDAR checklist

### Data availability

Cryo-EM maps are available from the Electron Microscopy Data Bank under accession codes EMDB-17209 (C3b-ISG65 composite map), EMDB-17219 (locally aligned CUB-TED-ISG65), EMDB-17220 (locally aligned C3c region), EMDB-17221 (C3b only) and EMDB-17273 (consensus map), while coordinates for C3b-ISG65 are available from the Protein Data Bank under accession code 8OVB.

The following datasets were generated:

| Author(s) | Year | Dataset title | Dataset URL | Database and Identifier |
|---|---|---|---|---|
| Cook AD, Higgins MK | 2024 | Human Complement C3b in complex with *Trypanosoma brucei* ISG65 | https://www.rcsb.org/structure/8OVB | RCSB Protein Data Bank, 8OVB |
| Cook AD, Higgins MK | 2024 | Human Complement C3b in complex with *Trypanosoma brucei* ISG65 | https://www.ebi.ac.uk/emdb/EMD-17209 | EMDataBank, EMD-17209 |
| Cook AD, Higgins MK | 2024 | Human Complement C3b in complex with *Trypanosoma brucei* ISG65 | https://www.ebi.ac.uk/emdb/EMD-17219 | EMDataBank, EMD-17219 |
| Cook AD, Higgins MK | 2024 | Human Complement C3b in complex with *Trypanosoma brucei* ISG65 | https://www.ebi.ac.uk/emdb/EMD-17220 | EMDataBank, EMD-17220 |
| Cook AD, Higgins MK | 2024 | Human Complement C3b | https://www.ebi.ac.uk/emdb/EMD-17221 | EMDataBank, EMD-17221 |
| Cook AD, Higgins MK | 2024 | Structure of complement C3 bound to *Trypanosoma brucei* ISG65 | https://www.ebi.ac.uk/emdb/EMD-17273 | EMDataBank, EMD-17273 |

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
