## [Editor Report · eLife assessment]

This **fundamental** study significantly advances our understanding of how parasites evade the host complement immune system. The new cryo-EM structure of the trypanosome receptor ISG65 bound to complement component C3b is highly **compelling** and well-supported by biochemical experiments. This work will be of broad interest to parasitologists, immunologists, and structural biologists.

---

## [Referee Report · Reviewer #1 (Public Review)]

The authors set out to use structural biology (cryo-em), SPR and complement convertase assays to understand the mechanism(s) by which ISG65 dampens the cytotoxicity/cellular clearance to/of trypanosmes opsonised with C3b by the innate immune system.

The cryo-EM structure adds significantly the the author's previous crystallographic data because the latter was limited to the C3d sub-domain of C3b. Further, the in vitro convertase assay adds an additional functional dimension to this study.

The authors have achieved their aims and the results support their conclusions.

The role of complement in immunity to *T. brucei* (or lack thereof) has been a significant question in molecular parasitology for over 30 years. The identification of ISG65 as the C3 receptor and now this study providing mechanistic insights represents a major advance in the field.

The authors have appropriately put their results into perspective with other recent reports on the role of ISG65.

---

## [Referee Report · Reviewer #3 (Public Review)]

The authors investigate the mechanisms by which ISG65 and C3 recognize and interact with each other. The major strength is the identification of exo-site by determining the cryoEM structure of the complex, which suggests new intervention strategies. This is a solid body of work that has an important impact in parasitology, immunology, and structural biology.

Comments on revised version:

The authors have addressed all the previous concerns.

---

## [Author Response]

The following is the authors’ response to the original reviews.

**eLife assessment**
This important study advances our knowledge of how parasites evade the host complement immune system. The new cryo-EM structure of the trypanosome receptor ISG65 bound to complement component C3b is highly compelling and well-supported by biochemical experiments. This work will be of broad interest to parasitologists, immunologist, and structural biologists.

We thank the reviewers and editorial team for this assessment of our work.

**Public Reviews:**

**Reviewer #1 (Public Review):**
The authors set out to use structural biology (cryo-EM), surface plasmon resonance, and complement convertase assays to understand the mechanism(s) by which ISG65 dampens the cytoxicity/cellular clearance to/of trypanosomes opsonised with C3b by the innate immune system.The cryo-EM structure adds significantly to the author's previous crystallographic data because the latter was limited to the C3d sub-domain of C3b. Further, the in vitro convertase assay adds an additional functional dimension to this study.The authors have achieved their aims and the results support their conclusions.The role of complement in immunity to *T. brucei* (or lack thereof) has been a significant question in molecular parasitology for over 30 years. The identification of ISG65 as the C3 receptor and now this study providing mechanistic insights represents a major advance in the field.
**Reviewer #2 (Public Review):**
This is an excellent paper that uses structural work to determine the precise role of one of the few invariant proteins on the surface of the African trypanosome. This protein, ISG65, was recently determined to be a complement receptor and specifically a receptor of C3, whose binding to ISG65 led to resistance to complement-mediated lysis. But the molecular mechanism that underlies resistance was unknown.Here, through cryoEM studies, the authors reveal the interaction interface (two actually) between ISG65 and C3, and based on this, make inferences regarding downstream events in the complement cascade. Specifically, they suggest that ISG65 preferably binds the converted C3b (rather than the soluble C3). Moreover, while conversion to a C3bB complex is not blocked, the ability to bind complement receptors 1 and 3 is likely blocked.Of course, all this is work on proteins in isolation and the remaining question is - can this in fact happen on the membrane? The VSG-coated membrane is supposed to be incredibly dense (packed at the limits of physical density) and so it is unclear whether the interactions that are implied by the structural work can actually happen on the membrane of a live trypanosome. This is not necessarily a dig but it should be addressed in the manuscript perhaps as a caveat.

We thank the reviewer for their positive response our work. We fully agree with the reviewer about the caveats which come from this work being done in a biochemical context. We have addressed this in lines 223-24 and 327-333.

**Reviewer #3 (Public Review):**
The authors investigate the mechanisms by which ISG65 and C3 recognize and interact with each other. The major strength is the identification of eco-site by determining the cryoEM structure of the complex, which suggests new intervention strategies. This is a solid body of work that has an important impact on parasitology, immunology, and structural biology.
**Recommendations for the authors:**

**Reviewer #1 (Recommendations For The Authors):**
A paper by Sulzen et al was published online on 27th April in Nature Communications that has a similarity (the cryo-EM structure) to this paper. This does not detract from the value of this paper. The authors should, however, include a "compare and contrast" section in this paper to explain similarities and differences in the conclusions. For example, while this paper demonstrates that ISG65 does not prevent C3 convertase activity, the Sulzen paper suggests it does prevent C5 convertase activity. The compatibility of these conclusions should be discussed.

Two studies of ISG65 were published shortly after submission of this manuscript (Sulzen et al and Lorenzen et al) and we have added a brief comparison of the conclusions of these papers here. These mentions include lines 151, 155-6, 201-2, 274-278, 292-93 and 321-323. For a more in-depth comparison we have published an opinion piece in Trends in Parasitology, which discusses all three of these papers and which we also now reference here.

Could the authors comment as to whether they think the association of C3b with the unstructured region of ISG65 comes about via S-S shuffling? I.e., is C3B first thioester linked to VSG and then this rearranges to ISG65 through C3b-ISG65 proximity?

We thank the reviewer for the interesting suggestion. However, we are not aware of evidence showing that C3b, which has been conjugated to a target protein through its covalent ester bond, then becomes transferred to a second target protein. As ISG65 can bind to C3 as well as C3b, we think that the conjugate could form when ISG65-bound C3 converts to C3b, becomes reactive and, through proximity, is most likely to conjugate to ISG65. Whether this occurs to a substantial degree in trypanosomes, or whether it is more likely that ISG65 interacts with C3b which is already VSG-conjugated, requires further experiments. We have edited lines 217-222 to make this point more clearly.

**Reviewer #3 (Recommendations For The Authors):**
The authors previously reported that ISG65 C-terminus is so flexible and is not resolved in their 2022 ISG65-C3d (TED of C3b) crystal structure, which is the same case here in the cryo-EM structure of ISG65-C3b. Thus, I am wondering how C3b might find the flexible C-terminus and form a covalent bond.

We think that the answer to the reviewer’s question relates to local concentration. When two reactive compounds are not attached together, then they diffuse freely in three-dimensions and their likelihood of colliding and reacting is subject to the randomness of Brownian motion. However, if they bind together through an interaction distinct from the reactive residues, then this increases their relative local concentration and the likelihood of collision and reaction taking place. In the case of ISG65, this is coupled with the ability of ISG65 to bind to C3 before it converts to C3b and becomes reactive. The interaction of ISG65 with C3/C3b will therefore bring together the reactive residues and increases the probability that they will collide and form a conjugate. Our control with BSA, which does not bind to C3/C3b, and does not form these conjugates supports this conclusion. We have edited lines 217-222 to clarify.

I also find it puzzling that deleting L2 or L3 in ISG65, which they found forming additional contracts with CUB domain of C3b (12 times binding tighter), does not affect the ISG65-C3b conjugate formation in the in vitro C3 convertase formation assay.

When we consider the affinities that the L2 and L3 loop deletions variants have for ISG65, and the concentration of ISG65 in the C3 convertase assay, we would predict that the conjugates still form with the ΔL2 and ΔL3 variants. This binding would therefore increase the relative local concentration of the reactive residues and ensure preferential conjugate formation, as we observe.

(1) Page 2 bottom line, "In particular, loop 2 forms a direct contact with the CUB domain of ISG65, centered around an electrostatic", ISG65 should be C3b.

We thank the reviewer for spotting this. It has been corrected.

(2) Page 4, "We found that ISG65 does not complete with either factor B or Factor D and does not block the binding of factor Bb (Figure 3b). This suggests that the C3 convertase can form in the presence of ISG65", "complete" should be "compete".

It has been corrected.

(3) Page 4, "revealed that in the presence of ISG65 a high molecular weight band appeared, which we identified through mass spectrometry to be a conjugate of ISG65 with C3b". There is no mass spectrometry data in the manuscript to support this.

We agree with the reviewer that this data should be included in the paper and have now added it as Supplementary Table 3.

(4) Page 5, "By inhibiting binding of CR2 to C3d, ISG65 will reduce the likelihood that B-cell receptor binding to trypanosome antigens will result in B-cell activation and antibody production." - this sentence is a bit confusing.

We have clarified this point in lines 243-245.

(5) Related to Figure 2a. "This structure reveals the two distinct interfaces formed between ISG65 and C3b (Figure 2a)." It would be clearer to label where interface 1 and interface 2 are in Figure 2a.

We have now labelled interfaces 1 and 2 above the insets in Figure 2a.

(6) Related to Figure 2C. I suggest mutagenesis to validate ISG65 L2/L3 - C3b CUB domain interaction, i.e. mutate ISG65 (N188, R187, Y190) and perform SPR with C3b.

We agree with the reviewer that this experiment was a valuable validation of our structural data. To achieve this aim, we changed our SPR assay, coupling C3 variants to the chip surface in an orientation which would match their conjugation to a pathogen and allowing us to reliably compare the affinities of ISG65 variants. We then assessed the binding of ISG65, ISG65∆L2, and the ISG65L2N188A,H189A,Y190A proposed by the reviewer. As predicted from the structure, both loop 2 deletion and mutation reduced the affinity for C3b but did not affect the affinity for C3d, suggesting that the difference in affinity of ISG65 for C3b and C3d is due to the observed interface 2. This new data is described in lines 150-168 and is presented in Figure 2c.

(7) Related to Figure 3a. Is the C3b only structure in the presence of ISG65 the real C3b only? Discussion can be added.

Our cryoEM analysis of the ISG65-C3b mixture yielded three dimensional classes which contained clear density for ISG65 and those in which there was no density for ISG65. While the reviewer is technically correct, and we cannot be 100% sure that there is not an entirely disordered ISG65 attached to these ‘unbound’ C3b, we think that this is extremely unlikely. In either case, these ‘unbound’ C3b are indistinguishable from other structures of C3b and the argument in the paper stands. We have added a clause in lines 178-179 to make this point.

(8) Related to Figure 3e. There is no label for WT and deletion mutants. Also, L1 and L3 deletion does not seem to show on the gel.

We have added these labels.